# Resolving Long-Standing Uncertainty about the Clinical Efficacy of Transcutaneous Electrical Nerve Stimulation (TENS) to Relieve Pain: A Comprehensive Review of Factors Influencing Outcome

**DOI:** 10.3390/medicina57040378

**Published:** 2021-04-14

**Authors:** Mark I. Johnson

**Affiliations:** Centre for Pain Research, Leeds Beckett University, Leeds LS1 3HE, UK; M.Johnson@Leedsbeckett.ac.uk

**Keywords:** transcutaneous electrical nerve stimulation (TENS), pain, analgesia, neuromodulation, systematic review, meta-analysis, randomized controlled clinical trial

## Abstract

Pain is managed using a biopsychosocial approach and pharmacological and non-pharmacological treatments. Transcutaneous electrical nerve stimulation (TENS) is a technique whereby pulsed electrical currents are administered through the intact surface of the skin with the intention of alleviating pain, akin to ‘electrically rubbing pain away’. Despite over 50 years of published research, uncertainty about the clinical efficacy of TENS remains. The purpose of this comprehensive review is to critically appraise clinical research on TENS to inform future strategies to resolve the ‘efficacy-impasse’. The principles and practices of TENS are described to provide context for readers unfamiliar with TENS treatment. The findings of systematic reviews evaluating TENS are described from a historical perspective to provide context for a critical evaluation of factors influencing the outcomes of randomized controlled trials (RCTs); including sample populations, outcome measures, TENS techniques, and comparator interventions. Three possibilities are offered to resolve the impasse. Firstly, to conduct large multi-centered RCTs using an enriched enrolment with randomized withdrawal design, that incorporates a ‘run-in phase’ to screen for potential TENS responders and to optimise TENS treatment according to individual need. Secondly, to meta-analyze published RCT data, irrespective of type of pain, to determine whether TENS reduces the intensity of pain during stimulation, and to include a detailed assessment of levels of certainty and precision. Thirdly, to concede that it may be impossible to determine efficacy due to insurmountable methodological, logistical and financial challenges. The consequences to clinicians, policy makers and funders of this third scenario are discussed. I argue that patients will continue to use TENS irrespective of the views of clinicians, policy makers, funders or guideline panel recommendations, because TENS is readily available without prescription; TENS generates a pleasant sensory experience that is similar to easing pain using warming and cooling techniques; and technological developments such as smart wearable TENS devices will improve usability in the future. Thus, research is needed on how best to integrate TENS into existing pain management strategies by analyzing data of TENS usage by expert-patients in real-world settings.

## 1. Introduction

Pain is managed using a biopsychosocial approach, and pharmacological and non-pharmacological methods. Therapeutic neuromodulation techniques deliver thermal, mechanical, chemical and electrical stimuli to the body, and are recommended in medicine, physical therapy and nursing for relief of pain. Electrotherapeutic neuromodulation techniques are categorized as; invasive, such as percutaneous electrical nerve stimulation, electroacupuncture, spinal cord stimulation and deep brain stimulation; or non-invasive, such as transcutaneous electrical nerve stimulation (TENS) and TENS-like techniques.

Delivering electricity through the intact surface of the skin for therapeutic purposes is an age-old technique. The Ancient Egyptians placed electrogenic fish on the skin to discharge electricity on painful body parts to ‘numb’ the pain [1]. The invention of machines to generate electricity fueled the manufacture of electrotherapeutic devices in the late 1700s, although the development of pharmacological based therapies for anesthesia and analgesia meant that electrotherapy was never accepted within mainstream medicine. Interest in electrotherapy to alleviate pain (electroanalgesia) was rekindled in 1965 following the publication of *Pain mechanisms: a new theory* by Melzack and Wall which suggested that electrical stimulation of low threshold peripheral nerves in the skin could inhibit activity of centrally located nociceptive transmission neurons, thus relieving pain [2]. In 1967, Wall and Sweet reported the success of percutaneous electrical stimulation of low threshold afferents in the skin to alleviate chronic neuropathic pain [3] and Shealy et al., reported success of electrical stimulation of the dorsal columns to alleviate pain associated with cancer [4]. Non-invasive electrical stimulation of the skin using surface electrodes (i.e., TENS) was used to forecast success of invasive electroanalgesic techniques until Long et al. reported in the early 1970s, that TENS may be beneficial in its own right [5].

Throughout the 1970s, published clinical observations without control groups suggested that TENS alleviated pain, resulting in practitioners using TENS as an adjunct to core treatment. Throughout the 1980s, physiological research demonstrated that electrical stimulation of peripheral nerves inhibited ongoing transmission of central nociceptive neurons via segmental and extrasegmental mechanisms, with various neurochemicals involved including gamma-aminobutyric acid (GABA) and opioids [6]. Clinical research suggested that TENS reduced pain in patients, but there were relatively few randomized placebo-controlled trials to confirm that benefits were due to electrical currents per se. The 1990s saw the publication of the first systematic reviews of RCTs comparing TENS with placebo TENS (i.e., no currents). Many of these initial reviews were inconclusive raising doubt about the efficacy of TENS. Misgivings about offering TENS in clinical practice arose i.e., ‘Would clinically relevant reductions in pain still be achieved without batteries in the TENS device?’

Three decades have elapsed since the first systematic reviews, yet recent reviews and meta-analyses, including Cochrane reviews, remain inconclusive or conflicting, despite a constant stream of new RCTs. Recommendations from clinical guideline panels about whether to offer TENS are inconsistent, causing uncertainty for patients, practitioners and policy makers. In the U.K. the National Institute of Health Care Excellence (NICE), recommend that TENS should be offered as an adjunct for osteoarthritis [7] and rheumatoid arthritis [8], but not for non-specific chronic low back pain [9] or intrapartum care [10].

On 7 April 2021, NICE published clinical guidelines on the assessment of chronic pain and the management of chronic primary pain in over 16s [NG193] [11]. NICE did not recommend TENS for chronic primary pain due to insufficient available data. Only two trials were included in their meta-analysis.

The debate about the efficacy of TENS has remained unresolved despite over 50 years of published research. Analyses of the costs, risks and benefits suggests that TENS rates favourably with standard care and NICE thresholds for Quality-Adjusted Life Year [12,13,14]. Thus, it seems timely to re-examine why such a large amount of published research has failed to resolve the issue of whether TENS alleviates pain.

### Aim

The purpose of this comprehensive review is to critically appraise previous clinical research on TENS to inform future strategies to resolve the ‘efficacy-impasse’. A brief overview of the principles and practices of TENS is provided for readers unfamiliar with TENS treatment. This is followed by a description of systematic reviews of RCTs, and a critical appraisal of factors influencing findings; including sample populations, outcome measures, TENS techniques, and comparator interventions. Issues arising are used to optimise the design of future RCTs, and systematic review and meta-analysis. It is suggested that the ‘efficacy-impasse’ would be resolved by large multicentred RCTs using an enriched enrolment with randomized withdrawal design, and a meta-analysis of the effect of TENS on pain intensity during treatment, irrespective of pain condition [15]. Finally, consideration is given to the consequences of being unable to determine efficacy due to insurmountable methodological, logistical and financial challenges.

The review is narrative in style and draws on three decades of experience of conducting reviews and meta-analyses of TENS for various types of pain. Narrative reviews can be vulnerable to selection and evaluation biases and opinion-oriented arguments, so readers are directed to key references for comprehensive coverage of topics of further interest. The intention is to challenge dogma where necessary and to catalyse scholarly debate about future directions for research, practice and health care policy.

## 2. Principles and Practice of TENS

This section provides a brief overview of the principles and practice of TENS as a foundation for issues discussed in later parts of the review. Readers familiar with TENS may wish to proceed directly to Section 3.

### 2.1. Introduction

In health care, transcutaneous electrical nerve stimulation (TENS) refers to the use of a portable device that generates pulsed electrical currents that are delivered across the intact surface of the skin via conducting electrodes to stimulate peripheral nerves (Figure 1). TENS is primarily used for symptomatic relief of a variety of types of pain irrespective of origin (i.e., nociceptive, neuropathic and nociplastic) or setting (i.e., inpatient, outpatient and palliative [16,17,18]). TENS is also used to manage faecal and urinary incontinence, constipation, nausea and vomiting, xerostomia, peripheral ischaemia and Reynaud’s syndrome, dementia, stroke (neuromuscular condition and neglect), oedema, wound healing, tissue regeneration (e.g., nerve, soft tissue, skin, and bone), reduction of tissue necrosis, sleep, fatigue, depression, and coma (for review, see [19]). The use of TENS for these non-painful conditions is outside of the scope of this review.

The safety profile of TENS compares favourably against medication. Adverse events associated with TENS are minor and include erythema and itchiness beneath or around the electrodes, and a vasovagal response in some individuals, manifesting as nausea and dizziness. Contraindications are few. From a legal perspective, manufacturers recommend not to use TENS for patients with epilepsy, malignancy, deep vein thrombosis or an active electrical device implant (e.g., cardiac pacemaker, implantable cardioverter defibrillator, spinal cord stimulators). It is possible to use TENS in some of these patients following careful assessment and providing electrodes are placed at sites distant to the hazard (for review see [20]; for safety guidelines see [21]).

When used to alleviate pain, TENS is usually administered to produce a strong comfortable sensation (electrical paraesthesiae) within, or close to, the site of pain (i.e., conventional TENS or sensory TENS, Figure 2). The sensory experience during TENS should be a pleasant ‘tingling’ or pleasant ‘pins and needles’ sensation and this should act to ‘soothe’ pain in a manner akin to ‘rubbing pain away’. For most people, beneficial effects are maximal during stimulation whilst the person experiences TENS sensation; thus, patients are advised to administer conventional TENS whenever they need to alleviate their pain, which may involve intermittently applying TENS throughout the day. Patients should be trained to self-administer TENS as needed following assessment by a practitioner to ensure that TENS is an appropriate treatment [22].

Self-administering TENS empowers patients to take control of their pain management and removes the need for patients to be supervised or to have to travel to the clinic for treatment. TENS meets the requirements of an ideal self-administered treatment because there is minimal potential for harm, toxicity, overdose, abuse, or interactions with other treatments or lifestyle. TENS may be prescribed by health care practitioners and can also be purchased without the need for a medical prescription [23]. Decisions on whether to offer TENS to a patient are taken according to the professional judgement of the healthcare practitioner. TENS is offered for symptomatic relief i.e., to ‘soothe pain’ in the moment, and as a first step therapy in primary, secondary and tertiary care settings within a stepped care model of pain management. Thus, TENS should be considered similar to heat therapies (e.g., hot packs and warm water bottles) or cold therapies (e.g., cold packs). (Figure 3).

### 2.2. Physiological Principles of TENS

#### 2.2.1. Electrical and Physiological Coupling

TENS is a technique-based intervention to selectively activate peripheral nerve fibres to elicit physiological neuromodulation. The electrical characteristics of currents produced by the TENS device influence which population of nerve fibres is activated (Figure 4).

The amplitude of pulsed current is the critical characteristic to influence which axons are stimulated. For conventional TENS, the amplitude of current is titrated to stimulate low threshold large-diameter, non-nociceptive nerve fibres (e.g., mechanoreceptive A-beta fibres) without concurrent activation of higher threshold small-diameter nociceptive nerve fibres (A-delta and C fibres, Figure 5). This is recognized by the user of TENS as a non-painful TENS tingling sensation and achieved using conventional TENS techniques (sometimes termed sensory TENS).

The frequency (rate) of pulsed currents determines the rate of neuronal impulses and is limited by the absolute and relative refractory periods for the axon. For conventional TENS, pulse frequencies below 250 pulses per second (pps) are used as they produce a pleasant tingling sensation. In principle, pulses of current between 50 and 500 microseconds in duration (i.e., pulse duration (width)) are optimal to activate low threshold large-diameter fibres (A-beta) without concurrently activating high threshold small-diameter nociceptive fibres (A-delta and C). Commonly, TENS devices deliver biphasic pulse waveforms with zero net current flow (Figure 4) as this is claimed to reduce the risk of adverse skin reactions due to a build-up of ion concentrations beneath electrodes [25]. Nevertheless, some TENS devices deliver monophasic pulse waveforms without users complaining of adverse skin reactions. For monophasic waveforms the cathode, which activates the axonal membrane, is placed proximal to the anode. The patterns of pulsed current may be continuous, burst (for low frequency stimulation), modulated (e.g., modulated amplitude, or frequency, or pulse duration), or alternating (e.g., switching from one frequency to another, Figure 4). Burst pulse patterns are used to administer low frequency stimulation and modulated pulse patterns used to lessen the likelihood of nervous system habituation.

#### 2.2.2. TENS Techniques

Conventional TENS is the most common technique used in clinical practice. Other techniques such as acupuncture-like TENS are used in specific situations or for patients who have not responded to conventional TENS (Table 1).

There is a raft of ‘TENS-like techniques’ described in the literature, with manufacturers and advocates claiming specific roles and indications. Whether such techniques show superiority in efficacy, safety or utility over placebo, conventional TENS and/or other treatments is a matter for debate and lies outside the scope of this review (for review see [26]).

#### 2.2.3. Physiological Mechanisms of Analgesic Action

TENS modulates nociceptive input at peripheral (‘peripheral impulse blockade’), segmental (‘spinal gating’) and extrasegmental (‘descending inhibition’) sites. Stimulation of low threshold large-diameter non-noxious cutaneous afferents reduces activity and excitability in sensitised or non-sensitised central nociceptive transmission cells, in segments of somatic receptive fields related to the location of the electrodes [27,28,29,30,31,32,33] (Figure 5).

Delivery of higher amplitude currents stimulates high threshold cutaneous (A-delta) afferents within deeper tissue and produces long-term depression of central nociceptor cell activity persisting up to 2 h post-stimulation [34,35,36]. In addition, stimulation of high threshold cutaneous (A-delta) and deep tissue afferents, activates extrasegmental structures in the brainstem and mid brain that project neuronal pathways to spinal nocieptive transmission cells, inhibiting their activity [29,37,38]. Intense afferent input to the central nervous system has been described as ‘acupuncture-like’ TENS (AL-TENS) and is often achieved using low frequency currents to produce muscle twitching which generates activity in deep-afferents coding movements of body parts (Figure 6).

TENS also extinguishes ‘incoming’ orthodromic impulses conducted in peripheral afferents (i.e., peripheral blockade). Low intensity (conventional) TENS extinguishes impulses conducted in low threshold cutaneous afferents arising from non-noxious somatosensory receptors such as mechanoreceptors [39,40]. At higher intensities TENS will extinguish impulses conducted in low and high threshold cutaneous afferents (non-noxious and noxious respectively), TENS-induced activity in high threshold cutaneous afferents is likely to be uncomfortable for the TENS-user [41] (Figure 7).

The central inhibitory effects during TENS occur through a complex interplay between excitatory and inhibitory neurotransmitters and neuromodulators, influenced in part by the frequency of the pulsed current (for review see [6]). These neurochemicals include opioids [42,43,44,45], GABA [46], acetylcholine [47], noradrenaline [48,49], serotonin [47], aspartate and glutamate [50]. TENS inhibits up-regulation of substance P, N-methyl-D-aspartate receptor 1 (NMDA-1), and cytokines (interleukin-1β, interleukin-6, tumor necrosis factor alpha) [51,52]; and suppresses expression of p-extracellular signal–regulated kinase 1/2 and cyclooxygenase-2 in the dorsal horn [53]. TENS reduces blood levels of the pro-inflammatory cytokine interleukin-6 in individuals with pain [54].

Recently, evidence has emerged that the analgesic effects of TENS for knee pain associated with osteoarthritis is influenced by genetic variation within the catechol-O-methyltransferase (COMT) and endothelin receptor type A (EDNRA) genes [55]. In future, genotyping may enable tailoring of TENS interventions to an individual’s genetic characteristic.

It is possible that TENS indirectly reduces pain by changing other physiological processes including; increasing microperfusion which is beneficial for ischaemic tissue, arterial insufficiency and claudication due to arterial and neuropathic diseases; reducing heart rate and systolic and diastolic blood pressure; and influencing local reflex control of visceral functions to counteract detrimental effects associated with constipation and urinary retention (for review see [19]).

## 3. Long-Standing Uncertainty: Efficacy and Effectiveness

### 3.1. Context: TENS and Evidence-Based Medicine

Decisions to offer treatments to patients involves the integration of clinical experience and patient values, with the best available research information about efficacy and effectiveness (i.e., evidence-based health care). Clinical experience suggests that TENS may provide short-term relief of pain during or immediately after treatment for any type of acute or chronic pain. Pains with a restricted distribution may be easier to treat because it is possible to permeate TENS sensation throughout the painful area. However, clinical experience alone cannot distinguish effects attributed to the active ingredient of a treatment from non-specific effects associated with the act of administering a treatment including patient expectation of a treatment having effects, natural fluctuations in symptoms, reporting bias to please the practitioner, and contamination from concurrent treatment.

In clinical research, RCTs are used to measure ‘efficacy’, in terms of beneficial (and harmful) effects of the *active ingredient* of a treatment, evaluated under ideal circumstances and controlled conditions. In contrast, evaluations of ‘effectiveness’ involve measurement of beneficial and harmful effects of the *whole treatment package* when given in real-life healthcare practice, and whether this treatment package matters to the patient. Efficacy studies tend to be ‘explanatory’ in design, whereas effectiveness studies tend to be ‘pragmatic’ in design [56]. For TENS to be considered efficacious, it is necessary to establish that beneficial outcomes are attributable to the active ingredient of TENS, over and above the *act of administering* TENS.

From a scientific perspective, the electrical currents are the active ingredient of TENS, so clinical efficacy is derived by comparing effects achieved during active TENS, with effects achieved when receiving placebo TENS using a sham device. In other words, ‘Do you need batteries in the TENS device to alleviate pain?’ This is answered using RCTs comparing TENS with placebo TENS. Systematic reviews and meta-analyses of multiple RCTs are used to assess the consistency of the direction, magnitude and precision of effect. Confidence and certainty of effect size estimates are appraised using tools such as GRADE (Grading of Recommendations, Assessment, Development and Evaluation).

### 3.2. Current Status of Evidence on the Clinical Efficacy of TENS

A brief overview of clinical research evidence on TENS for acute and chronic pain is provided below (for detailed reviews see [57,58]).

#### 3.2.1. TENS and Acute Pain

In 1996, Carroll et al. published a systematic review that provided evidence that the magnitude of postoperative pain relief during TENS was not superior to control comparisons, in 15/17 RCTs [59]. In 2003, a meta-analysis of 21 RCTs found that the mean reduction in analgesic consumption after TENS was 26.5% (range −6 to 51%) more than placebo. The size of effect depended on optimal TENS technique, i.e., strong, sub noxious electrical stimulation at the site of pain [60]. Since then, meta-analyses have suggested that TENS is superior to placebo for post-operative pain control and analgesic sparing outcomes following thoracotomy [61,62], and total knee arthroplasty [63,64]. Guidelines from the Australian and New Zealand College of Anaesthetists and Faculty of Pain Medicine, recommend TENS for acute pain, including pain after thoracic surgery [65].

In 2016, a Cochrane review on 19 RCTs (1346 participants) provided tentative evidence that TENS reduced the intensity of acute pain when given as a stand-alone treatment. Acute pains included procedural pains during cervical laser surgery, venipuncture, and sigmoidoscopy; and pain associated with post-partum uterine contractions and rib fractures [66]. There is a paucity of evidence to determine whether TENS is beneficial for the management of episodes of acute pain during sickle cell disease [67], dysmenorrhoea [68] and angina [69].

There is widespread use of TENS for pain during the early stages of childbirth, yet systematic reviews find evidence of efficacy to be weak or inconclusive [70,71,72]. At present, NICE recommend that TENS should not be offered to women in established labour, although it may be beneficial in the early stages of labour [10,73].

#### 3.2.2. TENS and Chronic Pain

The earliest systematic reviews on TENS for chronic pain were published in the late 1990s and were generally inconclusive [74,75]. In 2020, an overview of eight Cochrane reviews on TENS for chronic pain included 51 RCTs (2895 participants) and was inconclusive [76]. There are many systematic reviews on specific chronic musculoskeletal pain conditions, including osteoarthritis [77,78,79], non-specific low back pain [80,81], non-specific neck pain [82], epicondylitis [83] and fibromyalgia [84]. Generally reviewers judge evidence to be inconclusive due to a paucity of high quality RCTs. A large meta-analysis that pooled data from a variety of different musculoskeletal conditions (29 RCTs, 32 comparisons), found a significant reduction in pain during TENS compared with control [85]. A comparison of the efficacy of 34 treatments for non-specific chronic low back pain, estimated that the mean pain reduction during TENS was between 10 and 20 percent and comparable to other treatments including muscle relaxants and NSAIDs [86].

The most recent Cochrane review on TENS for neuropathic pain was inconclusive [87], as are systematic reviews of specific conditions including; painful diabetic neuropathy [88,89,90], pain following amputation [91], spinal cord injuries [92,93,94], central pain associated with multiple sclerosis [95], limb spasticity associated with damage to the central nervous system [96], cancer-related pain [97], chronic headache [98], carpal tunnel syndrome [99], and migraine [100]. A comprehensive non-systematic review on TENS for neuropathic pain was positive [17].

### 3.3. Everlasting Doubt about Efficacy and Effectiveness

One of the first case series on the use of TENS to manage pain was published in 1974 by Long et al. who stated that “*In spite of the fact that a large number of patients have been treated, these results must still be considered preliminary. The long-term effect over several years of treatment remains to be analyzed. The possibility of placebo effect must be evaluated. Nevertheless, the initial success that we have gained to date suggests that cutaneous electrical stimulation will be a significant advance in our ability to treat chronic pain*” [5] p. 267.

Three decades later, in 2000, the first Cochrane review on TENS for chronic pain was published and concluded: “*There is insufficient evidence to draw any conclusions about the effectiveness of transcutaneous electrical nerve stimulation (TENS) for the treatment of chronic pain in adults … Large multi-centre randomised controlled trials of TENS in chronic pain are urgently needed.*” [75] p. 2.

Five decades later, in 2020, the authors of the first overview of Cochrane reviews on TENS for chronic pain stated: “*We were therefore unable to conclude with any confidence that, in people with chronic pain, TENS is harmful, or beneficial for pain control, disability, health-related quality of life, use of pain-relieving medicines, or global impression of change.*” [76] p. 2. “*Given the resources allocated to TENS for the treatment of chronic pain in many countries there is an urgent need to undertake large RCTs to examine its effectiveness.*” [76] p. 9.

It is shameful that the vast amount of research spanning nearly half a century has failed to resolve the issue of TENS efficacy, resulting in longstanding uncertainty about whether TENS should be offered to patients in public health systems or covered by private healthcare insurance (e.g., within the National Health Service in the U.K., or by the Center for Medicare Services in the USA, respectively). In fact, there have been long-standing unresolved debates about the analgesic efficacy of many non-pharmacological treatments including; complementary therapies (e.g., acupuncture), electrophysical agents (e.g., therapeutic ultrasound) and manual therapies (e.g., massage techniques). Hence, a critical appraisal of factors contributing to long-standing uncertainty about the efficacy of TENS will be generalizable to other non-pharmacological analgesic treatments.

## 4. Factors Influencing Evaluations of TENS

The design and execution of RCTs can introduce biases contributing to overestimation and underestimation of treatment effects. The systematic review by Carroll et al. [59] evaluating the effects of TENS on acute post-operative pain demonstrated that non-randomised controlled trials tend to favour TENS, with 17 of 19 controlled studies without randomisation concluding that TENS was beneficial, compared with only two of 17 controlled studies with randomisation. Bjordal et al. [60] argued that trials with ineffective TENS dosage may have contributed to negative outcomes in studies on post-operative pain based on a meta-analysis that found TENS reduced postoperative analgesic consumption when optimal TENS technique and dosage were considered.

Bennett et al. [101] demonstrated that suboptimal dosing and inappropriate outcome assessment were particularly prevalent in RCTs on TENS. Bennett et al. assessed data related to treatment allocation, application of TENS, and assessment of outcomes from 38 studies included in Cochrane reviews on TENS for acute, chronic, and cancer pain. They found that poor implementation fidelity was a significant source of bias, contributing to inconsistency in treatment effects. Bennett et al. developed criteria for judging directions of bias in published RCTs of TENS which could aid the design future clinical trials (see later).

Sluka et al. [102] appraised factors that influenced the findings of clinical research evaluating the efficacy of TENS, and concluded that there needed to be more careful scrutiny of study methodology and the appropriateness of TENS treatment including; the nature of clinical populations, outcome measurements, TENS technique and regimens, and concurrent medication. Sluka et al. suggested that in future investigators should consider adequate dosing of TENS, medication usage, timing of outcome measurements, outcomes measured, and the clinical population to be studied, and that this should be informed using physiological principles and evidence from basic science research about the mechanisms of action and time-effect profiles of TENS.

### 4.1. Sample Populations

#### 4.1.1. Types of Pain

The belief that TENS is more suitable for certain types of pain has been present since its introduction in the late 1970s, yet there is much inconsistency in standpoints taken by practitioners, manufacturers, researchers and patients. Originally it was thought that TENS would be more beneficial for pains that were; superficial (cutaneous) rather than deep seated (visceral), localized rather than diffuse, mild-to-moderate rather than moderate-to-severe, chronic rather than acute, and musculoskeletal rather than neuropathic.

Physiological principles suggest that TENS should be more successful for pain that is confined to a small area and superficial in nature, although this is not always the case. For example, a high quality multicentered randomized placebo-controlled trial by Dailey et al. [103] found TENS alleviated movement evoked pain and other distressing symptoms associated with fibromyalgia, a condition that involves widespread multisite deep-seated musculoskeletal pain. In fact, evidence from animal studies suggests that TENS induced activity in deeper afferents may be partly responsible for antinociceptive effects [104]. Likewise, TENS of low threshold peripheral afferents may be expected to exacerbate tactile allodynia because activity in low threshold afferents has been implicated in generating tactile allodynia. TENS alleviates pain in some patients with tactile allodynia and exacerbates pain in others; careful positioning of electrodes can overcome the latter problem.

There is cursory evidence that patients’ satisfaction with TENS is related to the origin of pain [105], although generally, clinical research has failed to detect strong and stable relationships between patient characteristics, pain condition, treatment parameters (except TENS intensity) and pain relief [106]. Thus, pain pathology should not be a barrier to trying TENS, providing contraindications and precautions have been evaluated.

#### 4.1.2. Clinical Heterogeneity

The overview of Cochrane reviews on TENS for chronic pain described earlier [76], included a descriptive analysis of 51 RCTs (2895 participants) from eight reviews and was unable to conclude with confidence whether TENS was beneficial or harmful when used to manage pain. Reviewers were reluctant to pool data from different types of pain because clinical heterogeneity may affect the precision of effect estimates. The quality of the eight reviews scored high on a checklist to assess multiple systematic reviews (A MeaSurement Tool to Assess systematic Reviews, AMSTAR), whereas the quality of individual RCTs was judged by reviewers to be low due to inadequate sample sizes and risks of bias [107].

In 2007, Johnson and Martinson [85] pooled data from over 30 RCTs and over 1000 participants with chronic musculoskeletal pain and found that TENS was superior to controls (mostly placebo) at reducing pain intensity. They were criticised by Novak and Nemeth for combining sample populations of chronic musculoskeletal pains with multiple pathophysiological mechanisms (e.g., rheumatoid arthritis, osteoarthritis, chronic low back pain, ankylosing spondylitis, and myofascial trigger points) [108]. Novak and Nemeth argued that TENS was more effective for certain subpopulations of chronic pain than others, without providing examples, and that clinical heterogeneity would hinder generalization of findings. In rebuttal, Johnson and Martinson argued that clinical heterogeneity reflects the realities of clinical pain medicine and that their meta-analysis provided evidence “*… that on average TENS is effective, although we cannot determine which (if any) aetiologies it does not work on.*” [109] p. 229. To date, there is no robust evidence that specific electrical characteristics of TENS are efficacious for subpopulations of pain patients (see discussion later).

It is common for systematic reviewers to argue that combining data from various pain states is inappropriate because of variability in pathophysiology and/or clinical presentations. Despite this, analgesic interventions are used across a variety of acute and chronic pain conditions independent of cause. This includes over the counter analgesic drugs such as paracetamol, non-steroidal anti-inflammatory drugs (NSAIDS) and codeine, and non-pharmacological interventions such as exercise, massage, electrophysical agents, hot and cold therapies and acupuncture. Pain experience results from a mosaic of biopsychosocial factors and sometimes nociceptive input has limited influence, even for similar conditions. Thus, systematic reviews of specific pain conditions are unlikely to have homogeneous samples of pain participants. For example, an assessment of the efficacy of TENS for the treatment of “*… well-defined painful neurologic disorders …*” conducted by the Therapeutics and Technology Assessment Subcommittee of the American Academy of Neurology [88] included low back pain, although whether low back pain should be considered a well-defined painful neurological disorder, even in the presence of radiculopathy, is a matter for debate [110].

In summary, variability in response to TENS between and within individuals in RCTs is influenced by a complex interaction between biological, psychological, sociocultural and environmental factors associated with the context of an individual’s pain experience. The argument that data from different types of pain should not be pooled seems unreasonable because

TENS provides symptomatic relief of pain via physiological neuromodulation mechanisms that are not unique to, or influenced by, different types of painThere is no robust evidence that TENS has curative effects that are specific to pathology, and/or medical diagnosis, and/or type of painThere are no robust predictors of response to TENS according to type of pain

#### 4.1.3. Sample Size

The majority of RCTs have fewer than 50 participants in the TENS treatment arm. Very few RCTs have more than 100 participants in the TENS treatment arm (exceptions include an RCT on labour pain [111] and an RCT on fibromyalgia [103]). There are instances of RCTs with large sample sizes allocating participants into multiple small sized intervention groups, compromising statistical power. There are also instances of investigators stating that sample size calculations have been performed, although detail is omitted from study reports. It is suspected that many of these estimates are for total number of participants rather than numbers needed in each trial arm.

In meta-analyses, pooled data from many RCTs increases statistical power at the expense of increased clinical heterogeneity. Meta analyses conducted on specific pain conditions tend to have small sample sizes, seriously undermining confidence in conclusions. For example, the Therapeutics and Technology Assessment Subcommittee of the American Academy of Neurology concluded that “*TENS is established as ineffective …*” for chronic low back pain, although this conclusion was based on two RCTs and 114 participants receiving TENS and 87 receiving sham TENS [88]. They also concluded that TENS was “*… probably effective …*” for painful diabetic neuropathy based on two RCTs and 31 participants receiving TENS and 24 receiving sham TENS [88]. The rebuttal by Johnson and Walsh summarized the nonsense “*It seems unreasonable that the effectiveness of TENS, and subsequent clinical recommendations, can be established from studies with so few participants*” [110] p. 314.

Moore et al. has reviewed research on the impact of sample size on review outcome and argues that trial arms with fewer than 200 participants in RCTs, or fewer than 500 participants in meta-analyses, are at a high risk of bias, seriously undermining confidence in findings [112,113]. This rule-of-thumb has been adopted by the Pain and Palliative Support group of the Cochrane Collaboration. To date, no single RCT has met this threshold for participants for a TENS trial arm, and only two meta analyses have exceeded this threshold for pooled data; both reported that TENS was superior to placebo, for chronic musculoskeletal pain [85] and post-operative analgesic consumption [60].

The inclusion of many RCTs with small sample sizes in meta-analysis contributes to statistical heterogeneity and imprecision in data for all pooled analyses due to the influence of confounders associated with:Limitations in RCT design–i.e., methodological risk of biasInconsistency of results–i.e., differences in estimates of effect across RCTs (unexplained heterogeneity)Indirectness of evidence–i.e., differences across RCTs populations, interventions, outcome measures and comparisonsImprecision–i.e., the position of the 95% CI in relation to no effectPublication bias–i.e., tendency for RCTs with significant results to be published over those that do not have significant results

The impact of these confounders on confidence in overall effect size estimates are considered using tools such as GRADE (Grading of Recommendations, Assessment, Development and Evaluations) that enable the certainty of an effect estimate to be downgraded.

Generally, systematic reviews on TENS have not assessed levels of evidence against these specific GRADE criteria. Cochrane reviews have judged evidence to be of low or very low certainty based on insufficient RCTs and limitations in study design. Quantitative analyses of publication bias are also rare, although a meta-analysis of the efficacy of TENS for management of central pain in people with multiple sclerosis by Sawant et al. [95] calculated a failsafe N of 18.4. This suggests that more than 18 studies showing a null effect would be necessary to negate the statistically significant medium sized effect for reductions in central neuropathic pain intensity in favour of TENS, when compared with control comparisons (SMD = −0.349 (95%CI −0.609, −0.089); *p* = 0.009).

### 4.2. Outcome Measures

RCTs have evaluated the efficacy of TENS on a variety of outcomes including pain severity (intensity), pain interference (Brief Pain Inventory), sensory and affective dimensions of pain (McGill Pain Questionnaire), pain-free range of motion, tenderness to pressure (pressure algometry), analgesic consumption, and condition specific measures (WOMAC, Roland Morris Disability Questionnaire) including Quality of Life. The majority of RCTs measure pain intensity as the primary outcome using continuous (e.g., visual analogue scales) or ordinal scales (e.g., numerical rating scales).

#### 4.2.1. Issues Associated with Pain Intensity as a Primary Outcome

Traditionally RCTs compare pain intensity post-treatment relative to baseline between TENS and a comparator (e.g., placebo) as the primary outcome, and measured using standardised tools such as visual analogue scales or numerical rating scales (e.g., 0 = No Pain and 10 cm (or 100 mm) = Worst Pain Imaginable). Critically, baseline pain needs to be greater than mild pain to prevent ‘floor effects’ associated with insufficient baseline pain to relieve.

Pain rating scales have the allure of precision, reliability and objectivity although they are relatively blunt instruments capturing gross judgements about the severity of a complex dynamic multimodal subjective experience. Operational variabilities markedly affect precision of intensity ratings in RCTs on TENS including whether rating relates: to pain at present or is recalled; to pain at a discrete moment in time or averaged over specified time periods; and to spontaneous pain (e.g., at rest, background pain, flare-up pain) or evoked pain (e.g., during movement or provoked during pressure algometry to assess ‘tenderness’). Pain ratings may be captured when TENS is switched on, immediately after TENS has been switched off, or sometime after TENS had been switched off. Should the patient focus attention on pain sensation or on TENS sensation, or neither?

Rating the intensity of pain during TENS is compromised by the interaction of competing sensations of pain and TENS paraesthesia, which is not a natural sensation. It is common for some patients to describe the benefits of TENS as ‘distraction from pain’ rather than reductions in pain intensity, suggesting that language and semantics may be contaminating precision [106,114,115,116]. Whether ‘distraction from pain’ should be considered different to ‘relief of pain’ is a matter for academic debate, although patients report both to be beneficial. It is likely that at least some trial participants rate pain intensity as a blunt estimation of ‘is TENS benefiting me?’ or ‘does TENS help?’ rather than pain intensity per se.

Often these important details are absent from TENS trial reports. It takes time and effort to orientate participants to the specifics of rating pain intensity as a one item construct that is distinct from ‘distraction’, ‘satisfaction’, or ‘overall benefit’. Nevertheless, participants in some studies have distinguished constructs for pain intensity and satisfaction. Under double blind conditions TENS was found to be superior to placebo for treatment satisfaction [111] and willingness to use TENS again [70,71] but not for pain intensity during the early stages of labour. At present, patient satisfaction of treatment is not accepted as a valid measure of efficacy.

A further concern relates to the appropriateness of using parametric tests of average scores of pain intensity ratings measured as continuous data. Measuring continuous data using pain intensity scales is most sensitive to detect effects but can be misleading because patients tend to report substantial or minimal reductions in pain. This generates U-shaped intensity data distributions across participant samples, and averaging breaks assumptions of normality [112]. Thus, responder analyses of the number of participants reporting reductions in pain intensity of at least 50% (substantial reduction) or at least 30% (moderate reduction) would be more appropriate but are not common in RCTs of TENS.

#### 4.2.2. Consideration of Other Types of Outcome Measures

Semi-structured interviews of experienced TENS users by Gladwell et al., has demonstrated that outcomes used in TENS studies do not match benefits reported by experienced TENS users [116]. Gladwell et al. found that experienced users of TENS adopt sophisticated strategies to achieve outcomes beyond pain relief. They use TENS to ‘distract from pain’ and to alleviate sensations of muscle tension and spasm resulting in indirect benefits such as medication reduction, enhanced function, psychological well-being, and enhanced rest and sleep [114,115,117]. Thus, TENS was reported to facilitate improvements in activities of daily living such as walking, exercising, using public transport, shopping, returning to work, and sleep.

Gladwell et al. argues that unidimensional pain intensity scales and condition specific tools (e.g., Brief Pain Inventory, McGill Pain Questionnaire, WOMAC, Roland and Morris Disability Questionnaire, SF36) lack sensitivity and specificity to assess the complex pattern of potential benefits important to individual patients who employ various strategies of TENS usage in different contexts. Thus, TENS should be considered as a complex intervention and evaluated using methodologies sensitive to an individual’s context of use. Gladwell et al. suggests that patient reported outcome measures (PROMS) with a ‘function-outcome’ focus would be suitable to capture the variety of outcomes associated with individuals tailoring TENS treatment to their personal needs [116].

#### 4.2.3. Contamination of Outcome Measures by Other Treatments

In 2020, Grøvle et al. argued that the use of rescue and concomitant analgesics in placebo-controlled trials of pharmacotherapy for neuropathic and for low back pain contaminate findings, thus hindering interpretation and compromising replication of study findings [118]. Such contamination is problematic when participants have access to analgesics as part of multimodal treatment or as rescue medication. Participants receiving TENS or placebo interventions can titrate analgesic consumption to achieve adequate pain relief resulting in no differences in pain intensity between groups [60]. Hence, Bjordal et al. measured analgesic intake as a primary outcome and found that TENS reduced postoperative analgesic demand when compared with placebo during the first 3 days after surgery, with a reduction of opioid-induced side effects, including nausea and sedation [60]. In contrast, systematic reviews evaluating TENS for labour pain have failed to detect differences between TENS and placebo for the need of additional analgesia [72,119].

Contamination is less likely in studies evaluating TENS in in-patient settings because it is easier to measure and control analgesic intake, compared with out-patient settings when TENS is self-administered, and participants can access over the counter medication. Studies in out-patient settings are reliant on participants to accurately document consumption of concomitant treatment using tools such as pain diaries.

#### 4.2.4. Measurement Timepoints

A variety of measurement schedules have been used in RCTs on TENS and some do not match optimal TENS treatment schedules. Most RCTs evaluate TENS following a single in-clinic treatment or during a relatively short course of treatment that lasts fewer than two weeks. Using appropriate time points to measure outcomes is of critical importance so that effects of a single TENS treatment are distinguished from cumulative effects associated with repeated TENS treatment, and long-term outcomes associated with resolution of the painful condition.

Commonly RCTs measure outcomes before and after TENS, and sometimes neglect measurements during TENS. As TENS is used for symptomatic relief, pain intensity should be measured during stimulation. During repeated TENS treatment, measurements should be taken regularly to assess cumulative effects, and these should be distinguished from follow-up data collected sometime after a course of treatment has finished.

In health care, credence is given to long-term outcome, although there is a scarcity of data on long-term follow-up at six weeks, three months, six months, and 12 months after a course of TENS treatment.

Interpreting follow-up data for TENS needs careful consideration. The primary function of TENS is short-term symptomatic relief of symptoms rather than long-term cure of pathology. Thus, good quality efficacy data for short-term symptomatic relief of pain is of primary importance. Nevertheless, symptomatic relief may facilitate longer term functional benefit. For example, by relieving movement-related pain, TENS may indirectly lessen fear-avoidance of movement, resulting in more physical activity which may facilitate resolution of pain, and no further need for TENS. Thus, long-term follow-up data on TENS needs to distinguish whether participants stop using TENS because of resolution of the painful condition (a favourable outcome) or treatment failure (an unfavorable outcome).

### 4.3. TENS Interventions

#### 4.3.1. Variability of Criterion for TENS

There are a wide variety of non-invasive electrical stimulation devices and techniques used in clinical practice including; Transcutaneous electric acupoint stimulation; Transcutaneous spinal electroanalgesia; Acupuncture-like stimulation delivered using a Codetron device; Supraorbital transcutaneous stimulation; Non-invasive interactive neurostimulation using an InterX5000 device H-wave therapy; Neuromuscular electrical stimulation; Interferential current therapy; 5 KHz sine wave currents; Microcurrent electrical stimulation; High voltage pulsed direct current; Frequency rhythmic electrical modulation; and Auto-targeted neurostimulation. Some of these techniques have been included in systematic reviews despite not being TENS.

For example, RCTs by Itoh et al. state that they evaluated TENS for knee osteoarthritis [120] and chronic non-specific low back pain [121], but on close inspection the characteristics of currents used were interferential therapy “… *a single-channel portable TENS unit (model HVF3000, OMRON Healthcare Co Ltd., Japan), which sends between two electrodes a premixed amplitude-modulated frequency of 122 Hz (beat frequency) generated by two medium frequency sinusoidal waves of 4.0 and 4.122 kHz (feed frequency).*” [121] p. 23. RCTs by Itoh et al., have been previously included in a Cochrane review evaluating TENS for osteoarthritis [77] and a non-Cochrane meta-analysis evaluating TENS for low back pain [80]. H-wave therapy, Action Potential Simulation, Neuromuscular electrical stimulation, and Codetron have been categorized as TENS in systematic reviews.

Whether outcomes differ between TENS-like techniques and conventional TENS is a matter for debate. It has been suggested that interferential current therapy is no different to conventional TENS when administered to produce a strong comfortable electrical paraesthesia [122,123]. Tabasam et al. found that physiotherapists applied interferential currents in much the same way as conventional TENS for pain management [124]. In 2018, Almeida et al. [125] conducted a meta-analysis of eight studies with a pooled sample of 825 participants that found no difference in improvements in pain and functional outcomes between TENS and interferential current therapy.

#### 4.3.2. Variability of Electrical Characteristics of TENS

The belief that the electrical characteristics of pulsed currents used during TENS has a major influence on outcome, is longstanding. Manufacturers of standard TENS devices promote the use of pre-set combinations of electrical characteristics for different types of pain. A wide variety of pulse frequencies, pulse durations, pulse patterns and pulse amplitudes have been used in RCTs. Sometimes electrical characteristics are determined by the researcher and sometimes by the participants; sometimes electrical characteristics are fixed throughout treatment and sometimes they are adjusted according to the needs of the patient. The impact of variability in electrical characteristics on RCT findings is difficult to assess. Systematic reviews by Claydon et al. failed to detect consistent dose-related effects of specific electrical characteristics of TENS on chronic pain patients or healthy individuals exposed to experimentally induced pain [126,127].

Evidence from basic science and clinical research demonstrates that currents of sufficient amplitude are necessary for meaningful outcomes from TENS, i.e., a strong comfortable TENS sensation at the site of pain [60,101,128,129,130,131]. Some patients report that TENS sensations fade within treatment sessions as a result of the nervous system habituating to the repetitive non-noxious electrical pulses of TENS [132]. So, current amplitude should be adjusted to maintain this intensity if TENS sensation fades during treatment [133].

Basic science research has found that pulse frequency influences neurophysiological processes. Studies utilising animal models of nociception demonstrate that low-frequency TENS acts via mu opioid receptor whereas high-frequency TENS involves delta opioid receptors [43,44,45]. It is unclear whether such effects translate into different outcomes in humans, or that particular frequencies are beneficial for different types of pain. A systematic review of ten studies found no difference in hypoalgesia between pulse frequencies during conventional TENS [134]. However, evidence suggests that people with chronic pain are less likely to respond to low frequency TENS, which is mediated via mu-opioid receptors, if they also have opioid tolerance associated with long-term opioid medication acting on mu opioid receptors [43,44,135,136].

There is no robust evidence that subtle changes in waveforms, pulse frequencies, pulse durations or pulse patterns have significant and generalisable effects on clinical outcome, so patients are advised to adjust these characteristics on a moment to moment basis. Some patients have strong preferences for certain TENS sensations, and this may be related to the quality of their pain sensations [137]. Experienced TENS users report that the electrical characteristics of some TENS devices may be limited in choice, uncomfortable or too weak [114]. Evidence suggests that skin impedance may not be a critical factor in hypoalgesia during TENS [138], despite attempts to develop TENS devices with electrode arrays that can selectively deliver stimulation to sites with low impedance [139].

Overall, participant-reported strong but comfortable TENS sensations should be considered optimal for TENS. This would include using current amplitudes above motor threshold providing TENS was administered using a standard TENS device with the primary intention of stimulating peripheral nerves to alleviate pain. A comparison of TENS delivered at optimal (strong) versus sub-optimal (faint or barely perceptible) intensities would be valuable in future systematic reviews.

#### 4.3.3. Variability of Electrode Position: Site of Stimulation

There is variability in the site of electrode positions used in RCTs that have been included in systematic reviews. Physiological principles suggest that optimal effects of TENS will occur when electrodes are located at the site of pain or over nerve bundles proximal (or near) to the site of pain. These sites are commonly used in RCTs. In addition, electrodes have been positioned at contralateral body sites when the site of pain is hyposensitive (e.g., pain in the presence of numbness) or hypersensitive (e.g., sensitive post-amputation stump); and internally using a probe electrode (e.g., intravaginal [140,141,142] or intra-oral [143]).

There are many RCTs that have administered TENS to acupuncture points remote to the site of pain. Various electrical characteristics have been used including a technique described as transcutaneous electric acupoint stimulation (TEAS, TAES) whereby pulsed currents described as ‘dense-disperse’ are delivered using frequencies alternating between 2 pps and 100 pps. Commonly, transcutaneous electric acupoint stimulation is administered as a one-off treatment before surgery (i.e., pre-emptive) for post-surgical pain, although transcutaneous electric acupoint stimulation has also been administered post-operatively and/or to regional acupuncture points.

Whether transcutaneous electric acupoint stimulation should be included in systematic reviews of TENS is a matter for debate. Unclear terminology is a problem. For example, an RCT protocol published by Liang et al. [144] stated that “*Transcutaneous electrical acupoint stimulation (TEAS), which is also known as acupuncture-like transcutaneous electrical nerve stimulation (TENS) has been widely used in acute or chronic pain*.” [144] p. 1. Some opinion leaders would not consider TEAS similar to acupuncture-like TENS because of differences in electrical characteristics. Liang et al. [144] used a HANS Acupuncture Point Nerve Stimulator (HANS-100, Huawei Co., Ltd., Beijing, China), delivering a dilatational square wave current output at 2 Hz and 100 Hz alternative frequency (pulse width: 0.6 ms/0.2 ms) and 8–12 mA, and square-shaped electrodes (3 × 3 cm) to Ll4 (He Gu) and PC6 (Nei Guan), ST36 (Zu San Li) and SP6 (San Yin Jiao) for 30 mins twice a day. Acupuncture-like TENS is often described in vague terms as high intensity and low frequency pulsed electrical currents, sometimes at acupuncture points and sometimes not [145]. The term acu-TENS has been used to describe the application of TENS on acupoints, irrespective of the electrical characteristics of currents [146,147]. Clearly, there is an urgent need to standardize TENS nomenclature.

#### 4.3.4. Variability in Dose and Regimen

In 1997, McQuay et al. [148] published a systematic review of 37 RCTs on TENS for chronic pain that found a lack of evidence on which to judge effect. McQuay et al. concluded: “*The issue is one of dose. Many, perhaps most, chronic pain physicians who use TENS prescribe at least 30 min use twice a day for at least a month before any effect may be felt. This pragmatism is supported by the important study of Nash and colleagues,* [149] *who demonstrated a clear improvement in analgesic effects of TENS in a large number of chronic pain patients over a long period. None of the RCTs [in our systematic review] used doses of TENS which approached this. Duration of treatment was less than 4 weeks in 83% of the trials, and in 85% of the trials stimulation occurred less than 10 h per week, with 67% of the patients having less than ten sessions of TENS*.” [148] p. 48.

Two decades later, there continues to be an absence of RCTs evaluating long-term TENS treatment. Most RCTs evaluate TENS treatment over a period of days (e.g., post-operative pain) or weeks (e.g., chronic pain) rather than months. Many RCTs to date prescribe a fixed number of treatments of certain duration, whereas in clinical practice patients use TENS as often as needed. There remains a necessity to match treatment schedules with measurement protocols in RCTs, including distinguishing during treatment effects from cumulative effects and long-term outcomes (discussed previously). The need for pragmatic trials where participants are trained to personalize TENS treatment according to their needs is discussed later.

Repeated TENS treatment may result in a decline in pain relief in some individuals (TENS tolerance). A variety of biopsychosocial factors have been implicated including; dead batteries, perished leads, a worsening pain problem and waning of the initial enthusiasm for a new treatment. Repeated TENS has been shown to cause tolerance in mu and delta opioid receptors [150,151] with involvement of cholecystokinin and NMDA receptors [150,152,153,154]. The effects of low but not high-frequency TENS diminishes in morphine-tolerant rats [136]. Revisiting patient expectation of TENS, experimenting with electrode placements or the electrical characteristics of TENS (e.g., using alternating (4 pps and 100 pps) or mixed frequencies (4 pps on one day, 100 pps on the next day)) are potential solutions [151,155,156].

Paradoxically, repeated TENS treatment may intensify pain relief in others (cumulative benefit) with a variety of biopsychosocial factors involved including; increasing familiarity and competence at using equipment, improved skills at optimising treatment on an as needed basis, re-calibrating treatment expectations, and TENS indirectly improving functions of daily living. Repeated TENS also reduces neuronal excitability and sensitization at peripheral and/or central sites over time and may reset physiological sensitization to normal [6,31]. RCTs need to have flexibility in design to troubleshoot initial and/or declining response, and to prevent inappropriate categorization of participants as non-responders.

Investigators face major challenges in defining the scope of TENS in relation to type of devices, electrical characteristics, electrode placement sites, and adequacy of TENS technique. The following operational criteria have been published to define the scope of TENS by Johnson et al. [15]:Non-invasive electrical stimulation of the skin with the intention of stimulating peripheral nerves to alleviate painEquipment consisting of(a)‘standard TENS device’ defined as “*… a portable, battery-powered generator of monophasic or biphasic pulsed electrical current delivered in a repetitive manner, with a maximum peak-to-peak amplitude of approximately 60 milliamperes (mA) into a 1 kilohm load.*” [157] p. 12, and regardless of the device manufacturer; and(b)electrodes attached to the surface of the skin that would also include electrodes integrated into garments such as knee braces, cuffs, gloves and/or socksTechniques that produce strong but comfortable TENS sensations using any type of pulse pattern, pulse frequencies no more than 250 pulses per second (pps), and pulse durations no more than 1 millisecond (1000 µs). Intensities above motor threshold are acceptable providing TENS is administered with the intention of alleviating pain.Techniques that administer TENS at the site of pain or over nerve bundles near to the site of pain. TENS at remote acupuncture points would not be in-scope.Any TENS treatment schedule providing it matches with intended symptomatic relief of pain, and in RCTs matches measurement schedules.

### 4.4. Comparator Interventions

RCTs have evaluated TENS versus placebo TENS (e.g., sham TENS device without current); versus no treatment or waiting list control; versus treatments partly or wholly as standard of care (routine clinical practice); and versus other treatments, both pharmacological and non-pharmacological, not considered to be used in routine practice. Debates about the efficacy of TENS tend to concentrate on evaluations using placebo comparators.

#### 4.4.1. Placebo Controls for TENS

Placebo controls are used to isolate effects associated with the active ingredient of a real treatment from biases associated with receiving a treatment. In drug trials placebos are designed to be identical in appearance, taste and odour to the ‘real’ drug but contain an inert substance rather than an active chemical ingredient. It is important that placebo controls are indistinguishable from test interventions to conceal (blind) which intervention is placebo and which is ‘real’ from trial participants, practitioners and assessors. A variety of placebo comparisons have been used in RCTs of TENS including:Sham TENS devices with no current outputSham TENS devices with current above sensory detection threshold that fades to zero current output usually within 45 sActive TENS below sensory detection threshold, so the participant cannot sense itActive TENS above sensory detection threshold with infrequent pulses using interpulse intervals beyond that expected to produce physiological effects

Sham TENS devices with no current output are most commonly used as they isolate effects of pulsed electrical currents per se, i.e., the active ingredient of TENS.

Electrical currents are only a means (stimulus) to selectively activate low threshold afferents. Thus, from a physiological perspective any modality that selectively stimulates low threshold peripheral nerves, such as vibration, warmth and cold may generate similar effects as TENS, although the quality of the sensory experience may differ. The role of TENS sensation in outcome has been neglected. Historically, electrical paraesthesia was considered critical to the success of TENS and spinal cord stimulation. More recently the role of electrical paraesthesia in spinal cord stimulation has been revisited due to the development of spinal cord stimulators that do not generate electrical paraesthesia (i.e., HF-10). Moreover, a variety of non-invasive electroanalgesic (TENS-like) techniques that do not generate sensations during stimulation are available on the market (e.g., microcurrent therapy). The current consensus is that for conventional TENS, a strong comfortable TENS sensation is a criterion for adequacy of TENS. This raises challenges in blinding TENS interventions in RCTs.

#### 4.4.2. Blinding Placebo TENS Interventions

Blinding involves concealing the nature of interventions, including whether an intervention is a placebo. This is to reduce bias associated with expecting treatments to be beneficial and/or harmful from study participants and/or investigators and/or therapists. Double-blinding whereby participants and assessors are unaware of the nature of an intervention, is considered gold-standard in drug trials, although technique-based treatments that are administered by a practitioner may need to be triple blind, i.e., participant, practitioner and assessor. Leakage of blinding is reduced by isolating these individuals from each other, although this is particularly challenging especially between the practitioners administering and participants receiving treatments.

There is a longstanding debate about the fidelity of blinding participants and practitioners in studies of TENS. Active and placebo drugs can be made identical in appearance, with neither providing sensory cues. It is not possible to blind participants to TENS sensation, raising concern about trial participants guessing which intervention is the placebo (i.e., no current with no sensation). It is possible to create uncertainty about which intervention is ‘real’ using pre-study briefings. For example, by informing participants that:‘Some types of electrical stimulating devices produce tingling sensations during treatment, and some do not, such as microcurrent therapy’;‘During the study you may or may not experience sensations from the stimulating device’; and‘During the study you may or may not receive a placebo intervention’.

These types of statements meet ethical standards for informed consent because they are truthful whilst creating uncertainty about which intervention is a placebo.

Transient sham TENS devices that deliver current for 45 s that then fades to zero milliamps (i.e., no current) have been developed to reinforce blinding of participants, practitioners and assessors. Rakel et al. [158] found no differences in participant blinding between no current and transient current sham devices in a study of 69 healthy adults. Blinding and instances of leakage can be monitored using questions related to the credibility of the intervention such as ‘Do you think you received an active or placebo treatment?’ [103,158] or ‘Do you believe your TENS unit was functioning properly?’ [159]. Unfortunately, formally monitoring blinding is rare in RCTs on TENS.

There is inconsistency in judgements of the risk of performance and assessor bias (blinding) between systematic reviews. Some reviewers assign high risk of performance bias to all RCTs on the premise that it is impossible to blind the sensory experience of active TENS, whereas others assign low risk of bias on the premise that sham TENS devices coupled with participant briefing information create sufficient uncertainty. The majority of reviewers assign unclear risk of performance and assessor bias because RCT reports provide insufficient operational detail about blinding.

### 4.5. Absence of Evidence for Adverse Events

Critical appraisals of literature undertaken to develop safety guidelines suggest that TENS is a safe intervention and that adverse events are predominantly skin irritation and post TENS tenderness that are infrequent, mild in severity, and of minor consequence [20,21]. There is an absence of systematic review and meta-analysis evidence. Generally, RCTs evaluating TENS do not formally measure adverse events but document the occurrence of adverse events opportunistically [160]. Study reports often do not distinguish adverse events related to TENS with those related to other aspects of the study, such as medical procedures or treatments or general worsening of a medical condition. In 2020, Travers et al. [107] recommended in future, RCTs evaluating TENS should pre-specify formal procedures for documenting adverse events.

It is clear from Section 4 that there are a variety of shortcomings in RCTs evaluating TENS and this has contributed to inconsistency of findings of systematic reviews. Calls to improve the quality of RCTs evaluating TENS have been ignored for decades.

## 5. Resolving the Impasse

In 1997, McQuay et al. published a Health Technology Assessment of outpatient services for pain control in the U.K. [148]. McQuay et al. concluded “*TENS is of no value in acute pain*.” and “*The use of TENS in chronic pain may well be justified but it has not been seen.”* [148] p. 49. McQuay et al. stated that “*There is a requirement for a randomised trial to address the issue. It will be difficult to design and organise, it will need to be multicentred in the UK and other European countries may need to be included, it will require large numbers of patients and simple outcome measures. Without it, a potentially valuable intervention may be underused, or a useless intervention may continue in use*.” [148] p. 49.

There have been hundreds of clinical trials and over 100 systematic reviews published on TENS since McQuay et al.’s statement. In 2019, Gibson et al. [76] published an overview of Cochrane reviews and concluded “*Issues with quality, study size and lack of data meant we were unable to draw any conclusion on TENS-associated harms or side-effects or the effect of TENS on disability, health-related quality of life, use of pain-relieving medicines or people’s impression of how much TENS changed their condition.*” [76] p. 3.

To overcome the ‘efficacy-impasse’ there needs to be improvements in the quality of RCTs and a re-appraisal of approaches to systematic review. I offer two solutions:A large multicentered randomized controlled trial that enables participants to tailor TENS treatment according to their needs, including skills to optimise response and troubleshoot issues. Ideally, data should be gathered in a ‘real-world setting’ providing an ecologically valid insight to the value or otherwise of TENS in clinical practice. Enriched enrolment with randomized withdrawal studies are ideally suited for such needs.A comprehensive meta-analysis that estimates the magnitude of during treatment effects irrespective of the type of pain, and includes an evaluation of precision, consistency and certainty of the effect size estimate based on study quality and risk of bias.

### 5.1. Improving Future RCTs

In 2020, Travers et al. [107] recommended the following improvements for future research:Greater investment in large multicentred RCTs for precise estimates of effect size, and for scrutiny of large data sets not published in scientific literature such as those residing with device manufacturers.Better control of biases in RCTs, especially bias associated with blinding interventions and isolating individuals with different roles from each other.Improved clarity and detail in trial reports to enable replicability of methodology and evaluation of reproducibility of findings, in line with the Template for Intervention Description and Replication (TIDieR) checklist [161], including specific details of all aspects of TENS technique, including instructions for use.Evaluation of TENS in ecologically valid settings (e.g., self-administered at home for chronic pain) using outcome measures meaningful to individual patients (e.g., pain, function and adverse events) with long-term follow-up outcome.

Gladwell et al. argues that evaluations of TENS should capture what patients use TENS for rather than what practitioners or investigators prescribe, i.e., participants should not be passive recipients of treatment schedules [114,115,116]. To date, RCTs have evaluated TENS as a simple intervention, with participants receiving little information about TENS technique. Participants are rarely given opportunities to personalise treatment. Commonly, outcomes are pre-determined by investigators as pain intensity, despite evidence that patients use TENS to achieve a variety of direct and indirect benefits [114,115,116]. Rarely, are volunteers screened for alignment with the potential utility of TENS, ability to troubleshoot problems, non-response to previous treatments, long-term opioid use, low self-efficacy, catastrophising and a lack of willingness or likely compliance with instructions.

Gladwell et al. argues that TENS is a complex intervention and that long-term users of TENS optimise benefits and minimise problems by learning, through trial and error, personalised treatment strategies appropriate for their personal needs [114,115,116]. They learn how to select efficacious electrode positions and electrical characteristics (pulse amplitude, frequency and pattern) on an as needed basis, according to the nature of their pain and the context in which TENS is being used. For example, using TENS for a single treatment during a brief surgical procedure (e.g., colonoscopy) differs substantially from using TENS regularly at home use for chronic musculoskeletal pain.

RCTs need to be developed to enable participants to tailor TENS treatment according to their personal situation. Operational considerations have been incorporated into criteria published by Bennett et al. [101] to optimise allocation, application and assessment in RCTs, and are summarized in Table 2.

Many logistical challenges of tailoring and optimising TENS treatment within RCTs could be overcome by including a ‘run-in’ phase that could:Screen volunteers for factors influencing treatment failureDeliver TENS training on how to use TENS safely and how to optimise treatmentInclude a ‘skills development’ period where participants practice using TENS, personalise the positioning of electrodes and the TENS settings, and refine a pattern of TENS usage through a systematic process of trial and error. This will develop skills to adapt treatment according to needDetermine, through a process of shared decision-making, outcomes that participants find meaningful and realistically achievableIdentify participants who have unresolvable adverse events, are uncompliant or decide that TENS is not appropriate treatment for their needs

### 5.2. Enriched Enrolment with Randomised Withdrawal Design

Enriched enrolment with randomised withdrawal trials have been used to evaluate the efficacy of drug medication by identifying likely responders in a clinical population prior to randomisation into treatment arms for assessment of efficacy. Enriched enrolment with randomised withdrawal trials involve an observational ‘run-in’ phase (often of two weeks) to optimise dosage and assess adverse effects, and an RCT that evaluates efficacy using participants most likely to respond (i.e., an enriched sample). There have been no attempts to deliver an enriched enrolment with randomised withdrawal trial of TENS for any painful condition.

Phase one of an enriched enrolment with randomised withdrawal study on TENS would optimise TENS treatment and identify participants most likely to respond. Phase two would determine whether electrical currents are responsible for benefits or harms associated with treatment, as represented in Figure 8.

#### 5.2.1. Phase One–Open Label Run-In

Phase one utilises an open-label design with all participants receiving TENS treatment. Participants would be trained to use TENS and would learn how to self-administer treatment and tailor TENS technique according to personal need. Participants unable to tolerate adverse effects, are non-compliant with instructions for TENS technique, or opt not to use the treatment in the future are identified as ‘non-responders’ and withdrawn during or at the end of the phase. Only, potential treatment responders are permitted to enroll into the randomised controlled trial in phase two.

Phase one would be used to develop skills to be able to optimise electrode positioning and electrical characteristics which may be awkward, inconvenient and time-consuming. Recent technological advances that may help to optimise treatment include; electrodes woven into clothing, smart TENS electrodes with algorithms for precise targeting of currents, interfacing of the TENS device with mobile technology, and data capture systems that can monitor TENS usage [162,163,164,165].

#### 5.2.2. Phase Two–Enriched RCT

Phase two uses an enriched sample of ‘responders’ to assess whether effects observed in phase one were due to the active ingredient of TENS (electrical currents) rather than expectation or natural improvement of the condition. The enriched sample of participants would be free from unresolvable adverse events, and competent to self-administer and optimise TENS treatment. This should reduce treatment heterogeneity.

The ability to conceal (blind) real and placebo interventions is a challenge because participants would have had prior exposure to TENS sensation in phase one. Strategies to overcome risk of performance bias were discussed previously.

One criticism of enhancement enriched randomized withdrawal trial is that findings have limited external validity because populations arriving at clinics consist of non-responders. However, data gathered in phase one of the trial provides valuable ‘real-world’ data including, estimates of the incidence of responders, adverse events, compliance, and treatment satisfaction. This will provide a wholistic picture of efficacy and adverse events. The financial cost of delivering a large fully powered multicentred-placebo controlled enhancement enriched randomized withdrawal study would be high.

### 5.3. Improving Future Systematic Reviews and Meta-Analyses

Patients, clinicians, policy makers and funders urgently need an estimate of the magnitude of symptomatic relief of pain during treatment irrespective of the type of pain. This would answer the question ‘Does TENS sensation alleviate (soothe) pain in the moment?’ i.e., immediate short-term pain relief. An evaluation of precision, consistency and certainty of the answer needs to be based on study quality and risk of bias.

In 2019, Johnson et al. published a protocol for a meta-analysis to calculate effect size estimates for both benefit (pain intensity) and for harm (adverse events) from RCT data versus placebo (Meta-TENS study) [15]. The protocol pre-specified a detailed analysis of risk of bias, inconsistency, imprecision, indirectness and publication bias according to GRADE guidelines. The Meta-TENS study will provide one of three possible outcomes; (i) evidence of benefit, (ii) evidence of no benefit, or (iii) insufficient evidence to judge. It will be interesting to see whether the outcome is accepted by research and clinical communities. It is conceivable that the ‘efficacy-impasse’ may remain, irrespective of the outcome of the Meta-TENS study. Therefore, it is important to consider the consequences of being unable to resolve the ‘efficacy impasse’ due to insurmountable methodological, logistical and financial challenges; or entrenched dogma.

## 6. Being Unable to Determine Efficacy

Throughout the decades, calls for large multicentred RCTs have been unmet. The largest RCTs to date were delivered on modest funding (often below £250,000 GBP) restricting the samples within trial arms to no more than approximately 100 participants (e.g., Chesterton et al. [166], Palmer et al. [167], Dailey et al. [103]). Clearly, funding priorities are an issue, and whether funders would prioritize an expensive large-scale TENS study remains doubtful.

To date, the largest, multicentred placebo controlled RCT compared TENS (n = 103), with placebo TENS (n = 99) and no treatment (n = 99) for women with fibromyalgia [103]. The RCT overcame all methodological challenges, except for trial arm sizes of at least 200 participants. Results demonstrated that administering TENS at home whilst undertaking activities for at least 2 h per day for 4 weeks, reduced movement-evoked pain and movement-evoked fatigue, with improvements in global impression of change.

NICE judged these findings to be insufficient evidence to support a recommendation to offer TENS for chronic primary pain, concluding that “*Limited evidence for TENS showed no clinically important difference compared with sham TENS and usual care across several outcomes at less than 3 months, and no longer term evidence was identified.*” [11] p. 29. This judgement appears to be based on long-term outcomes rather than in the moment relief of pain at rest or on movement. Thus, one high quality RCT with 100 participants per trial arm is not considered ‘sufficient’ evidence for NICE to recommend TENS, and it is unlikely that additional large RCTs will be produced in the near future.

Hence the ‘efficacy-impasse’ is likely to remain for the foreseeable future, resulting in reduced availability of TENS within public health systems (e.g., NHS in the UK) and a reluctance of private healthcare insurance systems to cover treatment costs (e.g., by the Center for Medicare Services in the USA). Researchers place great credence on efficacy, whereas evidence-based health care practice also considers clinical experience, including patient values underpinned by physiological plausibility. Observational research spanning half a century demonstrates that some patients rely on TENS and use it for many years, and research from basic sciences supports physiological plausibility (see Section 2). In addition, technological advances to improve usability will sustain interest in TENS.

Nowadays, electrodes are available that are woven into garments and interfaced with smart phone technology for more precise targeting of currents without the need to reposition electrodes. TENS usage data can be uploaded to the Cloud and machine learning software used to create personalized TENS treatment schedules. Analysis of large data sets of usage and outcomes from real-world settings may be a relatively cost-effective means of gathering ecologically valid data from new patients to resolve doubts about the utility and potential efficacy of TENS. Therefore, it is unlikely that uncertainty about efficacy will discourage patients from purchasing their own TENS equipment or discourage practitioners from indicating TENS in the future (Figure 9).

## 7. Conclusions

This comprehensive review appraises reasons for longstanding uncertainty about the efficacy of TENS that has persisted for over half a century. Inconsistent and imprecise effect size estimates found by RCTs result from the use of inadequate sample sizes, heterogeneous populations, muddled measurements of outcome, inappropriate TENS technique, dosage and regimen, and difficulties blinding placebo interventions. The review reveals tensions between using currents to create pleasant non-painful sensations for symptomatic relief of pain, and using specific combinations of electrical characteristics of currents to target pathology and mechanisms associated with different pain conditions. Enriched enrolment with randomised withdrawal studies and a comprehensive meta-analysis to determine whether strong nonpainful TENS alleviates (‘soothes’) pain ‘in the moment’ are offered to resolve uncertainty about efficacy.

However, it may be necessary to concede that it is impossible to generate sufficient evidence about efficacy because operational challenges are insurmountable. This situation is not unique to TENS. There is longstanding uncertainty about analgesic efficacy for most nonpharmacological technique-based analgesic interventions including acupuncture, electrophysical agents (heat, cold, ultrasound, pulsed-shortwave, low-level laser) and a wide variety of manual therapies. Consequently, practice remains based on local policy and dogma. Perhaps clinicians, policy makers, funders and researchers need to reconsider the value of persisting with the production and publication of so many RCTs that fail to answer questions about efficacy.

## Figures and Tables

**Figure 1 medicina-57-00378-f001:**
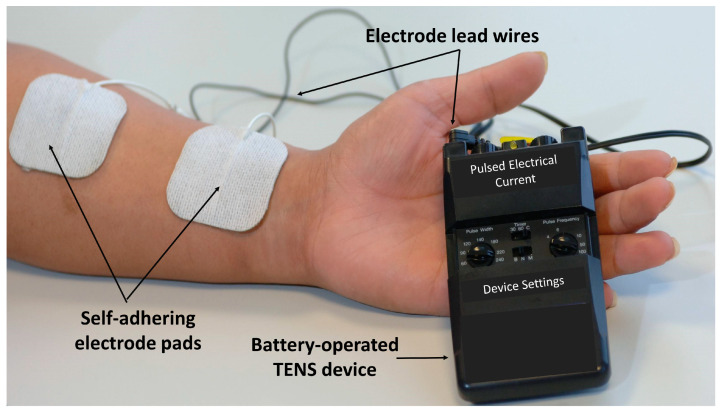
TENS equipment.

**Figure 2 medicina-57-00378-f002:**
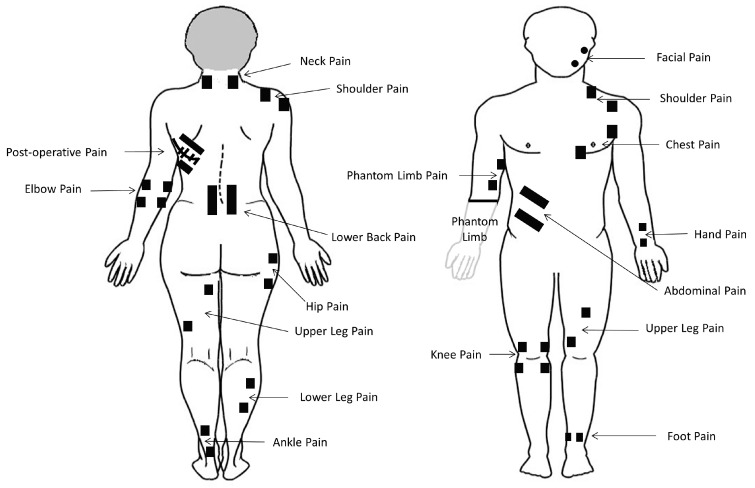
Common electrode positions for TENS. Black squares represent electrode pads.

**Figure 3 medicina-57-00378-f003:**
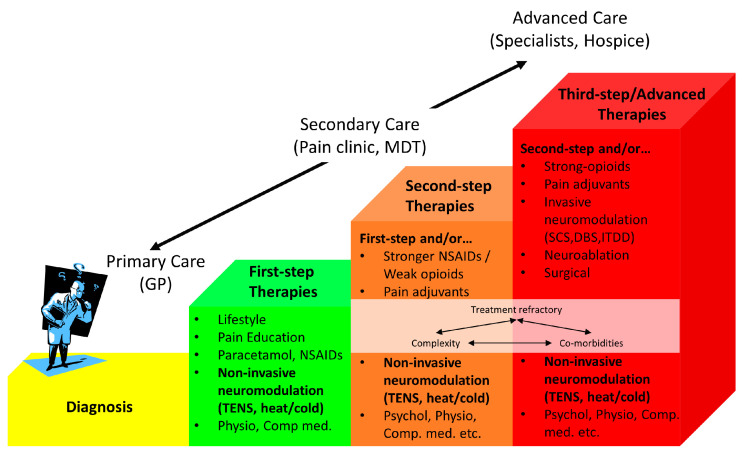
Stepped care model for pain treatment. Based on von Korff et al. [24]. Non-invasive neuromodulation techniques such as TENS are considered as adjuncts or as standalone treatment options at all steps of the care pathway. GP; General Practitioner: MDT; Multidisciplinary team: Psychol.; Psychology: Physio.; Physiotherapy: Comp. med.; Complementary medicine: NSAIDs; Non-Steroidal Anti-Inflammatory Drugs: SCS; Spinal Cord Stimulation: DBS; Deep Brain Stimulation: ITDD; Intrathecal Drug Delivery.

**Figure 4 medicina-57-00378-f004:**
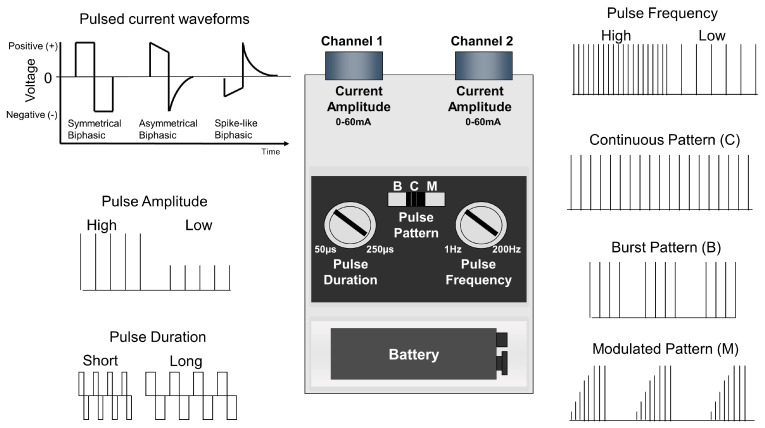
Output characteristics of a standard TENS device. Vertical lines represent a single pulse of current.

**Figure 5 medicina-57-00378-f005:**
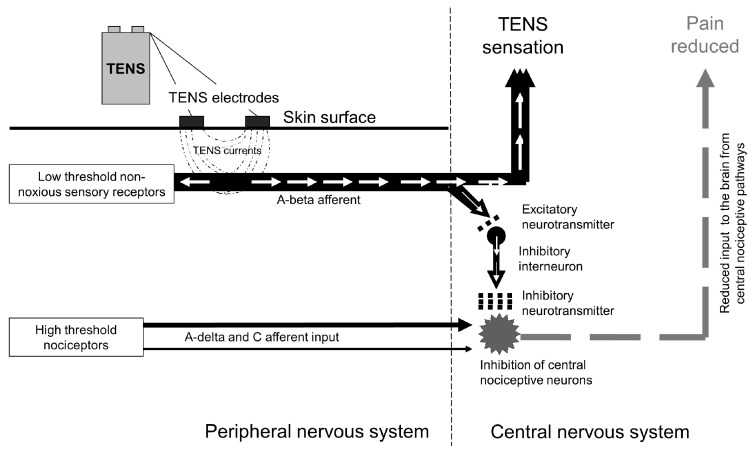
Selective activation of afferents using conventional TENS. The amplitude of currents is titrated to selectively activate low threshold nerve fibres (A-beta) generating nerve impulses (white arrows) that excite inhibitory interneurons in the central nervous system resulting in reductions in central nociceptive cell excitability and activity.

**Figure 6 medicina-57-00378-f006:**
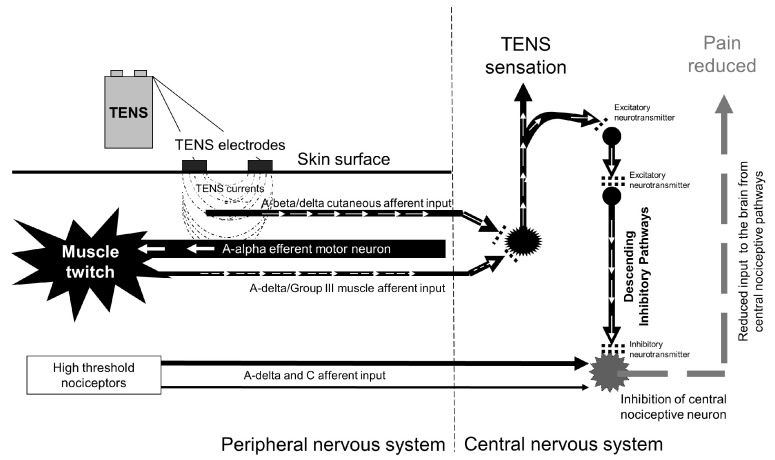
Activation of higher threshold afferents using acupuncture-like TENS. The amplitude of current is titrated to generate nerve impulses (white arrows) in A-beta and A-delta cutaneous afferents. In addition, if electrodes are positioned over motor nerves it is possible to generate nerve impulses in A-alpha motor neurons (white arrows) to elicit muscle twitching which in turn produces nerve impulses in A-delta/Group III muscle afferents (dashed white arrows). In both instances the peripheral input produces activity in descending inhibitory pathways arising in the brainstem and projecting to lower levels of the central nervous system (e.g., spinal cord), reducing central nociceptive cell excitability and activity.

**Figure 7 medicina-57-00378-f007:**
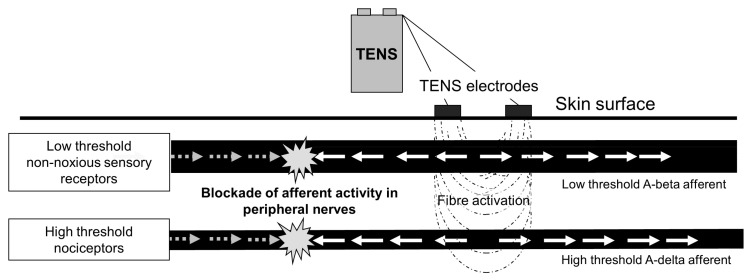
TENS induced blockade of afferent input from peripheral neurons. Impulses generated by TENS are conducted in both directions along the axon (white arrows). Those conducted toward the periphery (antidromic) extinguish nerve impulses arising from distal structures (dashed arrows) such as sensory receptors cells that are traveling in the normal direction (orthodromic). Nociceptive input conducted in higher threshold (A-delta and C-fibre) afferents is more likely to be blocked when higher amplitude currents of TENS are used to activate higher threshold axons (e.g., A-delta), but this is likely to produce uncomfortable TENS sensation that may not be tolerated by the patient.

**Figure 8 medicina-57-00378-f008:**
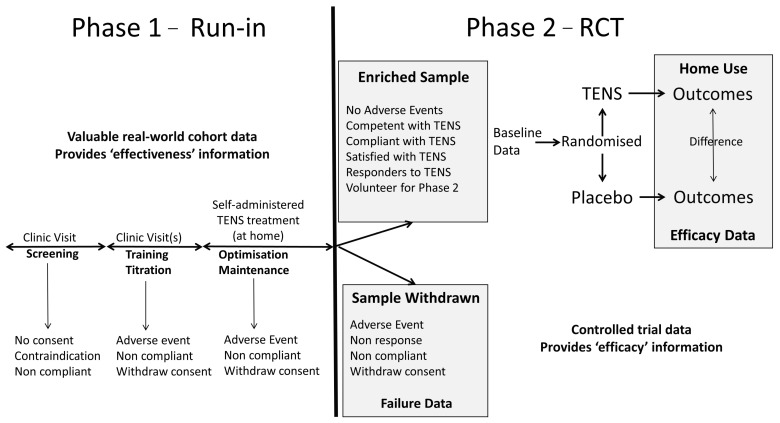
Enriched enrolment with randomised withdrawal study design for TENS. See text for explanation.

**Figure 9 medicina-57-00378-f009:**
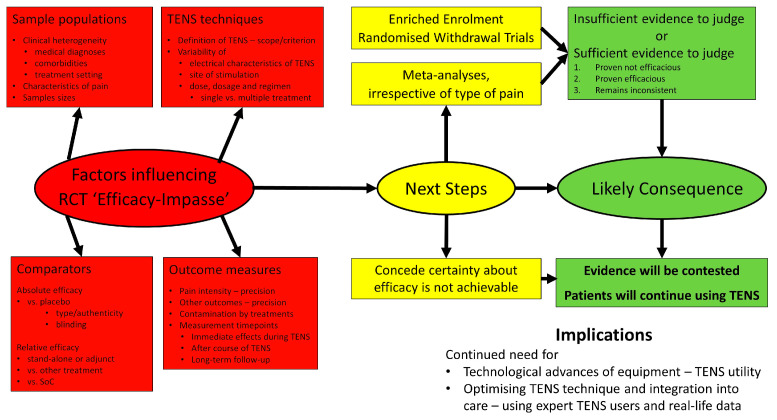
Summary of the factors contributing to the ‘efficacy-impasse’, next steps to resolve the impasse, and possible outcomes, consequences and implications going forward.

**Table 1 medicina-57-00378-t001:** Characteristics of common TENS techniques; pps = pulses per second.

	Clinical Purpose	Physiological Intention (Fibre-Type)	Desired Outcome-Patient Experience	Optimal Electrical Characteristics in First Instance	Electrode Position	Analgesic Profile for Most Patients	Duration of Treatment	Main Mechanism of Analgesic Action
Conventional TENS	TENS sensation soothes pain	Selective activation of low threshold non-noxious afferents e.g., arising from cutaneous mechanoreceptors (A-beta)	Strong comfortable electrical paraesthesia with minimal muscle activity	High frequency/Low intensity Current amplitude = varies according to patient Pulse pattern = continuous Pulse frequency = 10–200 pps Pulse duration = 50–500 (usually100–200) μs	Over site of pain Over main nerve bundle Dermatomal	Immediate relief Rapid offset of analgesia often within 30 min after TENS switched off	Whenever in pain-prn	Gating of peripheral nociceptive input-short acting neurotransmitters (segmental)
Acupuncture-like-TENS (AL-TENS)	TENS pulsing sensation and muscle twitching may be soothing or may be applied stronger as a counter irritant-accompanied by post-TENS relief of pain	(a) Activation of high threshold non-noxious/noxious cutaneous afferents (A-beta and A-delta) and/or (b) Activation of low threshold motor efferents to produce muscle twitching which generates impulses in afferents arising from muscle and deeper tissue (A-delta afferents)	(a) Strong comfortable pulsating sensation with/without (b) muscle twitching	Low frequency/High intensity Current amplitude =varies according to patient (a) Single pulse AL-TENS Pulse pattern = continuous Pulse frequency ≤ 10 pps Pulse duration = 200–500 μs (b) Burst pattern AL-TENS Pulse pattern = burst (frequency ~2 bursts per second) Pulse frequency = ~100 pps within burst Pulse duration = 100–200 μs	(a) Close to pain or over main nerve bundle if tolerated Dermatomal Sometimes placed on acupuncture point and described as ‘Acu-TENS’ (b) Over motor point/muscle at site of pain Myotomal	May be delayed onset of analgesia and maybe up to 30 min after TENS switched on Post-treatment relief may last >1 h after TENS switched off	Treatments of ~30 min a few times per day	Gating of peripheral nociceptive input - short acting neurotransmitters (segmental) and Activation of descending inhibitory pathways-long acting neuromodulators (extrasegmental)
Intense TENS	TENS sensation is a counter irritant	Generate nerve impulses in afferents arising from high threshold cutaneous afferents (A-delta)	Electrical paraesthesia that are slightly uncomfortable with minimal muscle contraction	High frequency/High intensity Current amplitude = discomfort but tolerable Pulse pattern = continuous Pulse frequency = ~50–200 pps Pulse duration > 500 μs	Remote body site as a counter irritant	Immediate action Post-treatment relief may last >1 h after TENS switched off	One off treatment for a few minutes during short ng painful procedures or breakthrough pain	Blockade of afferent input (peripheral) and Gating of peripheral nociceptive input - short acting neurotransmitters (segmental) and Activation of descending inhibitory pathways-long acting neuromodulators (extrasegmental)

**Table 2 medicina-57-00378-t002:** Operational considerations for the design of clinical trials for TENS. EERW = Enriched Enrolment with Randomisation Withdrawal.

Domain	Consideration	Operational Issues for EERW
**Population**		
Sample-size	Power calculation mandatory For most clinical outcomes, and especially pain, optimal sample size is ≥200 per treatment arm and minimal acceptable ≥50 per treatment arm	Pre-study–power calculation needed for phase 2 so sample enrolled in phase 1 would need to account for phase 1 participants ineligible to proceed to phase 2 (i.e., to account for dropout and withdrawal)
Sample-type	Any type of pain as no robust evidence that ‘pathology’ influences outcomes, especially those associated with pain. If sampling different types of pain consider the potential influence of pain context and treatment setting on outcome e.g., in-patient settings versus out-patient	Pre-study–this sample will be refined throughout phase 1 and some participants will be withdrawn (excluded) prior to phase 2 for failing to meet phase 2 eligibility criteria
**Allocation to intervention groups**		
Randomisation	Computer generated–independence from allocating investigators	Phase 2
Allocation concealment	Allocator should be independent from assessor and practitioner	Phase 2
**Application of interventions**		
Blinding-participants	It is not possible to blind sensory experience of TENS. Therefore, participants should be made uncertain which intervention is ‘real’ and which ‘placebo’ by using (a) real and sham devices that are identical in appearance, and (b) participant briefing instructions about how devices act to alleviate pain (see Calibration to study interventions). Assess whether participants believe (a) explanation about how different devices work is plausible, and (b) the device allocated was functioning properly	Phase 2
Blinding-practitioner	Practitioners should be uncertain which intervention is test and which control achieved by creating uncertainty about how devices act (see Calibration to study interventions). Assess whether the practitioner believes (a) the plausibility of the explanation about how different devices work, and (b) whether the device allocated to participant was functioning properly	Phase 2
Blinding–outcome assessor	Outcome assessors should be unaware which participant receive test and control interventions and be independent from other members of the investigating team	Phase 2
Blinding–data analyst	Statistical analysis should be conducted blind with analysists unaware whether data is test or control, and operating independently from other members of the investigating team until data collection is complete or if concern arises during the trial of excessive harm to participants through the occurrence of adverse events	Phase 2
**Assessment**		
Primary and secondary outcomes-type	Harmful effects Pre-specify a protocol to measure the incidence and severity of adverse events Consider adverse events as a primary outcome Beneficial effects Reliably measuring pain intensity is challenging and may not be appropriate as a primary outcome. Select outcomes that are more meaningful to participants, and ideally with specific objective goals easily verifiable with evidence other than subjective report (e.g., functional outcomes, consumption of analgesic medication, etc.). Consider use of patient-related outcome measures	Phase 1 and 2–in each phase outcomes will be a mix of pre-specified and negotiated with participant
Outcomes-meaningful to participant	Participants should negotiate outcomes that are meaningful to them in the context of their daily living including prioritising primary and secondary goals	Phase 2–outcomes that are meaningful to the participant are informed during phase 1 and then used as outcomes in phase 2
Measurement-timepoints	Measurement (a) during or immediately after the TENS intervention–single treatment effect (b) immediately at the end of a course of TENS treatment–cumulative effects over a course of TENS treatment (c) six months after the end of a course of treatment–long-term follow up	Phase 1 and 2–each phase will have different measurements, timepoints and endpoints
Measurement-analysis	Consider (a) continuous or dichotomous data e.g., for pain intensity use responder analyses (dichotomous data) in addition to averages (continuous data) (b) primary endpoint(s) for phase 2 (c) appropriate communication of the analysis in the trial report, allowing extraction of data for systematic reviews	
**Anticipation of intervention**		
Expectations-sensations associated with stimulation	Calibrate participants about sensations associated with stimulation using approaches and briefs designed to create uncertainty e.g., (a) ‘some types of electrotherapy do not produce sensations during stimulation (e.g., microcurrent electrotherapy)’; (b) ‘you may receive an intervention that does or does not produce a sensation during treatment’; and (c) ‘you may receive an intervention that does or does not deliver electric currents’	Phase 1 and phase 2. All participants would receive real TENS in phase 1 but briefings must not compromise blinding of participants and practitioners in phase 2
Expectations-treatment outcome	Calibrate participants to realistic treatment goals regarding TENS e.g., symptomatic relief of pain not curative. Participants should be made aware of direct and indirect outcomes associated with TENS including symptomatic relief of pain, and the relationship between dose, regimen and duration of effects	Informed by phase 1 and evaluated in phase 2-withdraw participant before phase 2 if incompetent
Expectations-compliance with treatment	Calibrate participants to the need to actively engage in (a) regularly self-administering treatment, (b) optimising treatment during each session, and (c) troubleshooting declining response	Phase 1–withdraw participant before phase 2 if incompetent
Expectations-Completion of knowledge and skills training	Calibrate participants to the need to engage in standardised training on (a) how to self-administer TENS or placebo interventions (b) how optimise their treatment and troubleshoot issues arising	Phase 1–withdraw participant before phase 2 if incompetent
Expectations-competency to self-administer TENS	Calibrate participants to the need to be competent to (a) self-administer TENS or placebo interventions, and (b) optimise their treatment and troubleshoot issues arising. This should be evaluated.	Phase 1–withdraw participant before phase 2 if incompetent

## Data Availability

Underlying research materials related to this paper (for example data, samples or models) can be accessed by contacting Mark I. Johnson.

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
