# Peer review of "Resolving Long-Standing Uncertainty about the Clinical Efficacy of Transcutaneous Electrical Nerve Stimulation (TENS) to Relieve Pain: A Comprehensive Review of Factors Influencing Outcome"

_medicina, 2021, doi:10.3390/medicina57040378_

Round 1

Reviewer 1 Report

The intention of this article is to "challenge dogma where necessary, to catalyse scholarly debate about future directions for research, practice and healthcare policy." I think the article is successful in its objectives, is very timely.

I have made reference to the pdf page numbers, not the manuscript page numbers:

Page 6 typo: monophasic is mis-spelt as monopahsic.

Page 9 typo: non-sensitised is spelt non-senstised.

Page 11: I think Carrol is spelt with double l, i.e. Carroll.

Page 12: this sentence needs to be revised: "Generally reviewers conclude that evidence inconclusive due to a of paucity of high quality RCTs."

Page 11: Again, I think Carrol is spelt with double l, i.e. Carroll.

Page 13: In this section, I think there is a word missing after "methodological", such as perhaps "methodological issues":
Sluka et al., [102] appraised factors that influenced the findings of clinical research evaluating the efficacy of TENS and concluded that there needed to be more careful scrutiny of study methodological and the appropriateness of TENS treatment including the nature of clinical populations, outcome measurements, TENS technique and regi-mens, and concurrent medication."

Page 14: Is there a missing word between "experience a" and "montage" here, such as "is"?: "Often pain experience a montage of biopsychosocial influences with specific pathophysiology unknown."

Page 16: Is the word "of" missing after allure: "Pain rating scales have the allure precision"?

Page 16: probalby need to insert "the" after patient here: "Should patient focus atten-tion on pain sensation or on TENS sensation, or neither?"

Page 17: the sentence "taking public transport to shops" shoudl be amended to "taking public transport, shopping..."

Page 17: the phrase "sleeping throughout the night without awakenings" goes beyond the findings of Gladwell et al, and should be shortened to "returning to work, and managing pain-related insomnia" or simply "and sleep".

Page 17: should the words "associated individuals" have a word in between, such as "with"?

Page 17: Please could the author review this sentence: "Most RCTs evaluate TENS following a single in-clinic treatment or during a relatively short course of treatment that lasts fewer than two weeks efficacy with a single treatment or up to a few weeks of treatment." Should there be a word or words between "weeks" and "efficacy"?

Page 20: this sentence needs to be revised: "Some opinion leaders would disagree that TEAS and AL-TENS because of differences in electrical characteristics"

Page 28: should the word "of" be between "magnitude" and "symptomatic"?

Author Response

Thank you for the time taken to review my manuscript and for your constructive comments that will improve the quality of the review. I agree with all suggestions and have identified my responses as bullets 

The intention of this article is to "challenge dogma where necessary, to catalyse scholarly debate about future directions for research, practice and healthcare policy." I think the article is successful in its objectives, is very timely.

I have made reference to the pdf page numbers, not the manuscript page numbers:

Page 6 typo: monophasic is mis-spelt as monophasic

  • Amended

Page 9 typo: non-sensitised is spelt non-senstised.

  • Amended

Page 11: I think Carrol is spelt with double l, i.e. Carroll.

  • Amended

Page 12: this sentence needs to be revised: "Generally reviewers conclude that evidence inconclusive due to a of paucity of high quality RCTs."

  • Amended

Page 11: Again, I think Carrol is spelt with double l, i.e. Carroll.

  • Amended

Page 13: In this section, I think there is a word missing after "methodological", such as perhaps "methodological issues":
Sluka et al., [102] appraised factors that influenced the findings of clinical research evaluating the efficacy of TENS and concluded that there needed to be more careful scrutiny of study methodological and the appropriateness of TENS treatment including the nature of clinical populations, outcome measurements, TENS technique and regi-mens, and concurrent medication."

  • Amended

Page 14: Is there a missing word between "experience a" and "montage" here, such as "is"?: "Often pain experience a montage of biopsychosocial influences with specific pathophysiology unknown."

  • Amended

Page 16: Is the word "of" missing after allure: "Pain rating scales have the allure precision"?

  • Amended

Page 16: probalby need to insert "the" after patient here: "Should patient focus atten-tion on pain sensation or on TENS sensation, or neither?"

  • Amended

Page 17: the sentence "taking public transport to shops" shoudl be amended to "taking public transport, shopping..."

  • Amended

Page 17: the phrase "sleeping throughout the night without awakenings" goes beyond the findings of Gladwell et al, and should be shortened to "returning to work, and managing pain-related insomnia" or simply "and sleep".

  • Amended

Page 17: should the words "associated individuals" have a word in between, such as "with"?

  • Amended

Page 17: Please could the author review this sentence: "Most RCTs evaluate TENS following a single in-clinic treatment or during a relatively short course of treatment that lasts fewer than two weeks efficacy with a single treatment or up to a few weeks of treatment." Should there be a word or words between "weeks" and "efficacy"?

  • Amended

Page 20: this sentence needs to be revised: "Some opinion leaders would disagree that TEAS and AL-TENS because of differences in electrical characteristics"

  • Amended

Page 28: should the word "of" be between "magnitude" and "symptomatic"?

  • Amended

Reviewer 2 Report

The author point out the need for resolving the uncertainty of the TENS efficacy to alleviate /or decrease pain.

This is a brillant idea, with a large amount of research and with, to date, no clear recommendations for (or against) the use of TENS in patient with pain.

However, several major issues appears that requires an extensive rewritting of the article. This includes:

  • The ms can be shortened and restructured so that essential information appears more easily to the naive reader. In this regard, please avoid the use of repetitive citation but instead summarize. The ms is unbalanced between a long part on the TENS effect that could be summurize (the author invites to skip this part for expert reader) ; identification of research issues, solution with a particular design and possible impasses.
  • I recommend to consider the presentation within paragraph : exposing the problem based on scientific paper, providing clinical recommendations and scientific recommendations.
  • The ms contains too many statement with no reference ; and too many sentence that remain too vague with no quantitative information that could be averaged data from scientific ms.
  • This includes the lack in characterising a pain condition (acute / chronic), the population, the type of current…
  • Possible impasses methodological, financial, logistical presented in the abstract should be revise in the text.

Author Response

Thank you for the time taken to review my manuscript and for your constructive comments that will improve the quality of the review. I agree with all suggestions and have identified my responses as bullets

Reviewer 2

The author point out the need for resolving the uncertainty of the TENS efficacy to alleviate /or decrease pain. This is a brillant idea, with a large amount of research and with, to date, no clear recommendations for (or against) the use of TENS in patient with pain.

  • Thank you for your support for the manuscript. As you can imagine a manuscript of this nature has not been easy to construct, so your constrictive comments are really appreciated.

However, several major issues appears that requires an extensive rewritting of the article. This includes:

The ms can be shortened and restructured so that essential information appears more easily to the naive reader. In this regard, please avoid the use of repetitive citation but instead summarize. The ms is unbalanced between a long part on the TENS effect that could be summurize (the author invites to skip this part for expert reader) ; identification of research issues, solution with a particular design and possible impasses.

  • I am Editor of the Special Issue on TENS and have produced a comprehensive review that will contextualise TENS for articles contained therein. I have edited the entire manuscript in line with your suggestions. The contextual information remains for lay readers, but I have flagged some additional parts of the manuscript that TENS specialists can ‘skip’ so that they are directed to salient points of my arguments.

I recommend to consider the presentation within paragraph : exposing the problem based on scientific paper, providing clinical recommendations and scientific recommendations.

  • As the other referees liked the structure I have decided to stay with the 5 section structure but have tried to improve the messages in these sections. I have emphasised these aspects of the paper more clearly within the relevant sections of the manuscript. In addition, I have added figure (Figure 9) to summarize the content of the manuscript

The ms contains too many statement with no reference ; and too many sentence that remain too vague with no quantitative information that could be averaged data from scientific ms.

This includes the lack in characterising a pain condition (acute / chronic), the population, the type of current…

  • I have added reference citations where appropriate, and emphasised more clearly when statements are conjecture-based

Possible impasses methodological, financial, logistical presented in the abstract should be revise in the text.

  • Amended as suggested

Reviewer 3 Report

This is a comprehensive review on decades of effort of evaluating the effectiveness of TENS in treating pain. The review is well organized and easy to follow. The author also provided his perspective for future investigations based on the findings. A minor suggestion, it would be great the content of the review can be summarized graphically illustrating the current status of TENS-related evaluations and the future research opportunities.

Author Response

  • This is an excellent suggestion
  • I have spent many hours pondering how best to present this visually and have inserted an additional figure - Figure 9. I hope it meets with your approval

Many thanks for taking the time to referee my manuscript

Round 2

Reviewer 2 Report

Dear author,

thank you for the revision of the manuscript. It appears that you did not take into account all of the comments made, in particular those in the pdf version of you initially submitted manuscript. I still consider that the manuscript could be improved. 

Yours sincerely.

Author Response

I appreciate your views and thank you for taking the time to consider my manuscript again. Unfortunately, the request in report two is extremely vague and does not respond directly to each of my original responses to report 1.

You say "It appears that you did not take into account all of the comments made, in particular those in the pdf version of you initially submitted manuscript". I am not entirely clear what you mean with respect to the pdf version and more importantly you have not specified which comments I have not taken into account

You say "I still consider that the manuscript could be improved", but you have not specified how? 

Can I re-emphasise that this is a 'comprehensive' review for specialists (hence it is contained with a special issue on TENS) and it is not an overview for lay readers. I have included context for the lay reader which is clearly flagged.

My understanding is that you believe that the manuscript should be 'extensively re-written' but you have not provided a clearly articulated framework on how to do this. 

Such a re-write will be against the wishes of the other reviewers.

As the world leader in this field of study and as Editor of the Medicina Special issue on TENS and TENS-like devices , I believe that the review should remain as presented. I have shared the manuscript with colleagues in the field of TENS research. They like the 5 section approach, the comprehensive nature of the arguments,  and they believe strongly that the manuscript should remain in its present form. They are looking forward to its publication.   

Thus, I am not prepared to compromise the quality of the paper by undertaking 'extensive re-writing' of the review. I re-iterate my original responses that I have flagged sections that can be 'skipped' by experts, have cited shorter summaries and supporting research on various related topics, and have added a figure to summarise material (at the request of reviewer three). As an author of over 250 research articles I believe that my use of in-text citations is appropriate for the size and scope of the review. 

I hope that this response meets with a favourable outcome

Kindest regards

Round 3

Reviewer 2 Report

I have no additional comment than the previously done and my decision remains unchanged (rejected).